# Nutritional Profile Analysis of Active and Sedentary Older Adults: Differences Between Spain and China

**DOI:** 10.3390/nu17071274

**Published:** 2025-04-06

**Authors:** Alba Niño, José Gerardo Villa-Vicente, Pilar S. Collado

**Affiliations:** 1Faculty of Physical Activity and Sports Sciences, University of León, 24007 León, Spain; 2Department of Physical Education and Sports, University of León, 24007 León, Spain; jg.villa@unileon.es (J.G.V.-V.); mpsanc@unileon.es (P.S.C.)

**Keywords:** aging, nutrition, physical activity, Tai Chi, Chinese

## Abstract

Background: the aging process is understood as a dynamic and plastic phenomenon involving various physiological and functional changes, which can be modified or modulated by external factors and lifestyle choices. Nutrition and regular physical activity have demonstrated their unique contribution to functional health and energy balance. This study investigates the impact of nutrition on physical condition among individuals over 65, focusing on energy intake, macronutrient consumption, and lipid profiles. Methods: for this purpose, four groups were analyzed: Maintenance Gymnastics, Tai Chi, Professional Tai Chi, and a sedentary control group. Objectives: the goal was to assess whether a better diet correlates with better physical outcomes. Results: all activity groups showed inadequate dietary patterns. The Spanish population consumed fewer carbohydrates and excess protein and fat, while the Chinese population adhered more closely to carbohydrate and lipid recommendations, maintaining a hyperproteic diet. Conclusions: the key differences were not in diet but in physical activity levels. The Asian lifestyle places strong emphasis on lifelong physical activity, complemented by culinary habits that enhance certain dietary parameters.

## 1. Introduction

The quest for a “fountain of youth” remains unfulfilled, as primary aging—the inevitable decline in cellular and biological function—cannot be prevented. However, secondary aging, influenced by lifestyle and disease, can be mitigated through interventions [1]. The WHO defines “active aging” as optimizing health, participation, and security to enhance quality of life, emphasizing social integration over passivity [2]. In Spain, individuals over 65 now exceed 19% of the population, with projections reaching 30% by 2050 [3,4]. China faces similar trends, expecting its 60+ population to double by 2040 [5]. As life expectancy rises, non-communicable diseases have replaced infectious diseases as leading causes of death, with obesity rates climbing, particularly between ages 35 and 74 [6,7]. These demographic shifts strain healthcare and social systems while increasing disease burdens [8]. Genetics influence only a third of life expectancy variance, with diet and lifestyle playing a predominant role in morbidity and mortality [9,10]. Global calorie consumption has risen since 1964, particularly in developing regions, exacerbating obesity and cardiovascular disease rates worldwide [11]. Based on all of the above, the objective was established to verify whether the nutritional requirements of older adults support the data on insufficiency and also present a possible relationship with physical activity. To this end, it was necessary to compare an Asian culture, where an active lifestyle predominates, with a Spanish culture, where a sedentary lifestyle predominates.

### 1.1. Nutrition of Older Adults

Older adults represent the fastest-growing population group worldwide, and with this increase comes a rise in nutrition-related problems [12]. Nutrition plays a crucial role in preventing and treating chronic diseases common in older age. However, many older adults experience malnutrition, which increases the risk of morbidity, mortality, and healthcare costs [13]. As people age, their energy needs decrease, and food intake significantly drops by about 25% between the ages of 40 and 70 [14,15]. This reduction in intake, combined with factors like a decreased appetite and limited access to food, leads to conditions such as sarcopenia, frailty, and functional decline [16,17]. Early identification and interventions are crucial in order to prevent a further deterioration in nutritional status. Even though the recommended daily allowance (RDA) can guide nutritional intake, it may not always meet the specific needs of older adults, as RDAs are typically based on younger, healthier populations [14]. Older adults often require more nutrient-dense foods due to age-related changes in nutrient absorption and utilization. A healthy diet for older adults, such as the Mediterranean diet (MD), is one that includes adequate amounts of energy and nutrients. The MD focuses on plant-based foods, olive oil, and a moderate consumption of animal products, and is linked to the prevention of cardiovascular diseases, diabetes, and cancer, and promoting longevity [18,19].

In Asia, particularly China, the approach to nutrition incorporates Traditional Chinese Medicine (TCM) alongside diet and physical activity. The typical Chinese diet is rich in vegetables, fruits, fish, and legumes, with rice as a central carbohydrate source. This diet also emphasizes plant-based proteins, such as soy and tofu, which contribute to better bone health and help prevent calcium loss, especially in menopausal women [20]. In general, the Chinese diet includes fewer animal-based foods, a lower fat intake, and more fiber compared to Western diets, reflecting a different approach to maintaining health and well-being in older age.

### 1.2. Alarming Data on Sedentary Behavior

Sedentary behavior is a growing global concern among older adults. A recent meta-analysis found a sedentary behavior prevalence of 31% and physical inactivity at 43% in this population [21]. Studies indicate that 60% of older adults sit for over 4 h daily, with 27% exceeding 6 h [22], and those aged 60+ spend 80% of their waking time sedentary [23]. High sedentary levels negatively impact cognitive function, depression, disability, and quality of life [24]. In Spain, 47% of the population does not engage in physical activity, ranking it ninth in the EU for sedentary behavior, with 19.4% of adults classified as sedentary (22.4% women, and 16.2% men). In contrast, China fosters active aging through community exercise traditions, potentially leading to lower sedentary levels. Additionally, physical activity is embedded in their philosophy of life, with individuals incorporating exercise into both work routines and leisure time.

### 1.3. Physical Activity in Older Adults

The WHO underscores the crucial role of physical activity in promoting healthy aging and quality of life. Its 2024 World Report on Aging and Health [25] confirms that regular physical activity benefits both physical and mental health, preventing non-communicable diseases and improving cognitive function, sleep, and mental well-being in older adults. However, 31% of adults and 80% of adolescents fail to meet the recommended activity levels, leading to a global goal of reducing inactivity by 10% by 2025 and 15% by 2030. Without intervention, public health systems could face costs of USD 300 billion between 2020 and 2030. While aerobic exercise improves cognitive function, it alone is insufficient for musculoskeletal health, necessitating a combination of strength, balance, and walking exercises to prevent frailty-related issues [26,27]. In Spain, maintenance gymnastics has been a primary activity for older adults, but Tai Chi (TC) has gained popularity due to its health benefits and meditative nature. Originating as a martial art, Tai Chi is now widely practiced worldwide, particularly in China, and is officially recognized by multiple sports organizations, including the Royal Spanish Federation of Judo and the International Wushu Federation. These organizations regulate all its activities.

### 1.4. Interaction Between Diet and Exercise in Older Adults

Several studies have examined the separate effects of increased physical exercise or dietary supplements on muscle mass and physical performance in older adults. However, less is known about the extent to which the benefits of training could be enhanced when both interventions are combined. Both external factors for improvement should be considered together when aiming to enhance the quality of life for older adults. Therefore, it remains important to continue investigating the beneficial, and likely unique, effects that calorie restriction and/or nutrient modification can provide, particularly for functionally frail older populations, as well as for physical activity interventions. Unlike Western culture, where older adults increasingly adopt more sedentary habits and consumption patterns outside dietary recommendations, Eastern behaviors across different ages show similarities in eating habits. In the case of older Chinese adults, their diets could be considered similar to those of the general population. Regarding physical activity, their lifestyles present even more marked differences, characterized by an active lifestyle in Asian societies compared to a nearly completely inactive one in the West. The Chinese philosophy of life goes hand in hand with the need for physical activity, specifically the practice of Tai Chi, and this is why their customs throughout life are shaped by these physical guidelines. Regardless of their work, they take necessary and practically mandatory breaks every day to engage in exercise. As a result, their eating schedules and daily activity routines differ significantly from those established in Spain.

## 2. Materials and Methods

An observational biometric study was conducted, with both descriptive cross-sectional and analytical retrospective cohort, based on an initial study by the same authors [28], which showed the functional capacity of older adults practicing Tai Chi, without considering its potential relationship with their diet. Therefore, this study involved the evaluation of food intake and nutritional calculation (specifically an analysis of energy intake, macronutrients, and lipid profile), with a 7-day dietary record [29], followed by the use of the DIAL^®^ software program (version 3.4.0.10) to obtain the nutritional data based on the amount of nutrients consumed. This program uses the recommended objectives provided by both the Consensus Document of the Spanish Society of Community Nutrition and the Food Composition Tables and the Institute of Medicine [30,31]. The study was designed according to the guidelines of the Declaration of Helsinki, and approved by the Institutional Review Board (or Ethics Committee) of León University (ETICA-ULE-004-2021).

### 2.1. Study Participants

This study provided the opportunity to analyze consolidated health-related habits (nutrition and physical activity) from two different cultures, conducted with older adults from Spain and China. A total of 113 individuals aged 65 years or older (71.53 ± 6.92 years) participated. Of these, 36 were practitioners of Maintenance Gymnastics (MG) (n = 30 women, n = 6 men), from the city of León and the town of Grado in Asturias, Spain. A second group consisted of 27 Tai Chi practitioners (n = 21 women, n = 6 men), residents of Madrid and Logroño, Spain. A third group comprised 27 professional Tai Chi athletes, who had a continuous practice in this sport and were Asian competitors in the Senior category (n = 12 women, n = 15 men), residing in the city of Shanghai, China. The fourth remaining group of 23 people corresponds to the sedentary control group (n = 14 women, n = 9 men), residing in Spain, who were not currently engaging in any physical activity nor had they practiced any in the last few decades. Both the Maintenance Gymnastics and Tai Chi groups had been practicing their respective sports for an average of ±4 years, with a weekly practice frequency of ±3 days. The professional Tai Chi practitioners from China had dedicated their entire lives to this sport, with ±12 h of practice per week, and were still competing at present.

### 2.2. Guidelines for Conducting Evaluations

The participants were informed through a meeting in their local area about the study’s evaluation and procedure, as well as the records and tests they were required to complete. The tests conducted included a 7-day consecutive dietary record of their usual food intake (with photographs taken of portion sizes) and completing all the tests that form part of the Senior Fitness Test. All participants signed the informed consent form to take part in the study and gave their consent for photographs to be taken. The dietary records were given to the participants during the initial briefing and setup of the study. Some were filled out with the assistance of the participants who required help, and they were also provided with a phone number for any questions or inquiries.

### 2.3. Exclusion and Inclusion Criteria

Inclusion criteria were as follows: being 65 years old or older, participating in supervised classes (Maintenance Gymnastics or Tai Chi) over 4 years prior to the study or not practising any kind of physical activity regarding the group considered sedentary. The exclusion criterion applied to those who suffered any severe illness or ailment which hinders mobility.

### 2.4. Evaluation Instruments

Each record required participants to note their food intake, supplements (vitamins, amino acids, etc.), drinks (alcoholic and non-alcoholic), and water consumed over the course of seven days. The time of meal start and finish, as well as the location of consumption (home, restaurant, or cafeteria), were recorded along with the overall menu, aiming to be as precise as possible. Additionally, the cooking method (boiled, roasted, fried, battered, etc.) and all possible details about the consumed foods (brand; type of oil; whole, semi-skimmed, or skimmed products; whole wheat bread, white, or sliced bread; type of cheese [aged, semi-aged, or fresh], etc.) were noted. Finally, the amount of food consumed was recorded as precisely as possible, using household measurements (glasses, cups, tablespoons, deep plate, flat plate, etc.). All calculations in the DIAL^®^ software program are based on a nutritional composition table of foods, which provides information about the average nutrient content and other substances provided by foods when consumed. The edible portions for each participant are expressed in grams per gram of whole food, while other information refers to every 100 g of the food’s edible portion. The program allows for modifications to ingredient amounts in each dish, as well as the inclusion of new dishes to simplify the program’s use and increase its versatility and adaptability for different populations, groups, age categories, or activities [32]. Table 1 and Table 2, compiled from the DIAL^®^ software, display the nutritional variables analyzed in the study.

Regarding the nutritional variables analyzed in the study, the RDA compliance for energy intake (Kcal/day), carbohydrates (g), proteins (% of RDA), energy availability (Kcal/kg of fat-free mass), lipids (g), and proteins (g) are presented as references. Regarding the lipid profile, the intake of SFA (g) (% of RDA in Kcal) is shown: intake of saturated fatty acids (SFAs) in grams, expressed as a percentage of the recommended caloric intake; MUFA (g) (% of RDA in Kcal): intake of monounsaturated fatty acids (MUFAs) in grams, expressed as a percentage of the recommended caloric intake; PUFA (g) (% of RDA in Kcal): intake of polyunsaturated fatty acids (PUFAs) in grams, expressed as a percentage of the recommended caloric intake; PUFA/SFA: polyunsaturated fatty acids to saturated fatty acids ratio; PUFA + MUFA/SFA: combined polyunsaturated and monounsaturated fatty acids to saturated fatty acids ratio; cholesterol (mg): cholesterol intake in milligrams; and cholesterol (mg/1000 Kcal): cholesterol intake adjusted for every 1000 Kcal consumed.

The Senior Fitness Test was administered to all participants, following an initial warm-up, before they began their usual physical activity to avoid the effects of fatigue and differences between groups due to their different exercises. In the days prior to the tests, participants were briefed on the tests and familiarized with them. Five tests were conducted, Leg Strength (LS), Arm Strength (AS), Leg Flexibility (LF), Arm Flexibility (AF), and Agility (Ag), from the validated test for older adults, the Senior Fitness Test, which allows for the assessment of physical condition [28].

### 2.5. Data Analysis

Quantitative variables are presented as mean values and standard deviations. In this study, comparisons of means were conducted using the appropriate statistical tests, employing IBM^®^ SPSS^®^ Statistics 25, version 25.0.0.0 (SPSS^®^ IBM Corp., Armonk, NY, USA). To evaluate ethnic differences between Spanish and Chinese participants, an independent-samples *t*-test was performed. To assess the impact of physical activity across all groups compared to the sedentary control group, an independent-samples *t*-test was also applied. Furthermore, to analyze differences among specific physical activity groups (sedentary, maintenance exercise, Tai Chi, and professional Tai Chi practitioners), a one-way analysis of variance (ANOVA) was conducted, provided that the assumption of normality was met across all groups. For variables that did not follow a normal distribution according to the Kolmogorov–Smirnov (K–S) test, a logarithmic transformation was applied to facilitate parametric comparisons. In post hoc analyses, since the assumption of homoscedasticity was not met in any case, the Welch’s statistic was computed, and the Games–Howell post hoc test was employed. For non-parametric post hoc comparisons, the Mann–Whitney U test was used.

## 3. Results

Table 1 shows the descriptive data among older people from Spain and China who practice Tai Chi, and Table 2 shows the descriptive data according to the type of physical activity in older people in Spain.

### 3.1. Differences in Energy Intake, Macronutrient Composition, and Lipid Profile Based on Physical Activity Type in Older Adults in Spain

Regarding the caloric intake and macronutrient distribution, Table 3 shows no statistically significant differences based on the type of physical activity practiced by Spanish older adults, nor in comparison with the sedentary group. All participants exhibited a low-carbohydrate and high-protein diet.

Without differences in the caloric profile of the diet according to the type of physical activity in the older Spanish population (Figure 1), it is observed that sedentary individuals, Maintenance Gymnastics practitioners, and Tai Chi practitioners reach 90.8%, 93.2%, and 79%, respectively, of the recommended %RDA. However, none of the groups meet the minimum recommended carbohydrate intake (only 41.6% of sedentary individuals and approximately 43% of those practicing Maintenance Gymnastics and Tai Chi), while they primarily exceed the protein intake, with a slightly lower excess in lipid intake.

### 3.2. Differences in Energy Intake, Macronutrients, and Caloric and Lipid Profile Between Older Adults from Spain and China Practicing Tai Chi

As shown in Table 4, there are statistically significant differences between older adults from Spain and China in carbohydrate consumption (39.88% higher in the Chinese group) and lipid consumption (67.52% lower in the Chinese group). Regarding the achievement of dietary goals, 92.6% of the Chinese participants meet the recommendation, compared to only 22.2% of the Spanish participants. Regarding the lipid intake, 40.7% of the Chinese sample meets the RDA, while 66.67% of the Spanish sample exceeds the recommendations. There are no significant differences in protein intake between the two ethnic groups, either in terms of its caloric percentage or its gram consumption, as both groups predominantly exceed the RDA and the respective objectives.

The differences shown in the caloric profile of the diet (Figure 2) correspond to a 19% higher compliance in carbohydrate consumption among older Chinese individuals compared to the Spanish, who show a deficient intake (only 43.19% of the %RDA). Additionally, there is a 17% higher lipid consumption in the older Spanish population compared to the Chinese, with the lipid intake being somewhat deficient in the older Chinese population, meeting only 19.8% of the %RDA.

Regarding the lipid profile of both groups, Table 5 shows a significantly lower intake of SFA in older Chinese individuals compared to Spaniards (74.9% lower in the Chinese group), with 74% of the Spanish group exceeding dietary recommendations, while 93% of the Asian sample meets them. The intake of MUFAs shows significant differences in the older Chinese population, with lower intakes than the Spanish (79.68% lower), although neither group meets the recommendations. For the PUFA intake, significantly lower values are observed in the older Chinese population compared to the Spanish (44.6% lower), although both groups exceed RDA recommendations. Regarding other indicators of lipid profile quality, in the PUFA/SFA ratio (>0.5), the Asian sample presents a significantly higher index compared to the Spanish group (24.32% higher). Additionally, in both this ratio (PUFA/SFA) and the PUFA + MUFA/SFA ratio (>2), a high percentage of the studied older Chinese population meets the targets. Regarding cholesterol intake (mg), the Asian sample shows a significantly higher consumption compared to the Spanish (24.51% higher), with no differences in relation to energy intake (mg/1000 kcal), although, in this case, the intake exceeds recommended levels.

Figure 3 shows significant differences between both ethnic groups in caloric percentage. In the older Spanish population, the percentage of energy derived from SFAs is 62.86% higher than the recommended target. Similarly, the percentage of energy from PUFAs exceeds the recommended target by 62% in the older Spanish population and by 19% in older Chinese individuals. In contrast, the caloric percentage from MUFAs is deficient in both ethnic groups, being 60.65% lower than the recommended target in Spaniards and 75.48% lower in Chinese individuals.

## 4. Discussion

The primary objective of this study was to determine whether the type of physical activity influences the dietary habits of older adults. The dietary patterns of individuals practicing two common physical activities among older Spanish populations—Maintenance Gymnastics and Tai Chi—were analyzed and compared with a sedentary control group.

Regarding energy and macronutrient intake, no significant differences were found among participants based on their physical activity levels. However, in terms of RDA compliance, sedentary individuals adhered more closely to recommendations, with an energy intake of 1708.8 ± 299.0 kcal/day, compared to 1820.1 ± 420.7 kcal/day in the Maintenance Gymnastics group and 1629.4 ± 355.0 kcal/day in the Tai Chi group. Despite having a lower intake, Tai Chi participants only achieved 79% RDA compliance, while sedentary individuals reached 90% and Maintenance Gymnastics participants 93%. Notably, all three groups exhibited a hyperproteic diet and excessive lipid intake, while the carbohydrate intake remained below recommended levels in 96% of sedentary participants, 89% of Maintenance Gymnastics practitioners, and 78% of Tai Chi participants.

These findings contrast with previous large-scale nutritional studies, which reported that more active older adults had a higher caloric intake, greater protein consumption, and higher carbohydrate intake compared to sedentary individuals [33]. Additionally, lipid profile results from those studies indicated that sedentary individuals consumed more fat, cholesterol, and saturated and polyunsaturated fatty acids than their active counterparts, which differs from our findings. Similar discrepancies were observed in studies comparing other physical activities, such as dance and walking, where increased hours of activity correlated with a greater energy intake [34]. However, this trend was not evident in the present study for Maintenance Gymnastics and Tai Chi.

Comparing these results with other populations, studies indicate that sedentary individuals often exhibit an imbalanced diet, characterized by a higher fat intake relative to carbohydrates and proteins [35]. Similarly, research on young athletes versus sedentary adults has shown that athletes’ nutritional status aligns more closely with the RDA, whereas sedentary individuals display a lower energy intake, a higher percentage of energy from fats, and lower micronutrient levels [36]. These differences were not observed in our sample of older adults. Other authors [37,38] have also reported findings consistent with ours, highlighting an excessive protein and fat intake alongside inadequate carbohydrate consumption among older individuals in Spain.

When considering different types of athletes, studies have found that triathletes, professional dancers, and skiers consume energy amounts equal to or above the RDA, whereas other groups, such as classical dancers and rhythmic gymnasts, exhibit lower energy intake [39]. Our findings align with the latter, as our physically active groups did not show an increased energy intake relative to the RDA. Furthermore, research indicates a negative association between low physical activity and overall health-promoting behaviors, including proper nutrition, stress management, and interpersonal support [40]. Other studies with older adults found that, while sedentary individuals showed a greater adherence to the Mediterranean diet, physically active individuals exhibited better overall fitness levels [41]. These insights suggest that older adults should aim to improve both their diet quality and physical activity levels for optimal health benefits.

Another objective of this study was to compare older Spanish individuals practicing Tai Chi with their Chinese counterparts, given the cultural significance of this activity in China, where it is practiced throughout life. Spain has undergone significant social and economic transformations in the 20th century, leading to changes in dietary patterns and lifestyle behaviors associated with improved socioeconomic conditions. However, these shifts have also contributed to increased obesity and related chronic diseases. In contrast, China has experienced rapid globalization-driven changes in food availability, dietary habits, and lifestyle, resulting in a rise in non-communicable diseases such as obesity and type 2 diabetes [42].

Despite the absence of significant differences, the Spanish population in our study reported a daily energy intake 10.75% lower than the Chinese (1629 vs. 1804 kcal/day), and in relative terms, 7.57% lower (25.79 kcal/kg body weight vs. 27.90 kcal/kg body weight). Macronutrient intake patterns were also distinct, with both populations exhibiting excessive protein consumption (77.66 g/day for Spanish vs. 80.37 g/day for Chinese). However, carbohydrate intake was significantly higher in the Chinese group, at 39.88% greater in absolute terms (160 g/day vs. 267 g/day) and 38.5% higher relative to body weight (2.54 g/kg vs. 4.13 g/kg). Only 22% of Spanish older adults met the carbohydrate intake recommendations, compared to 95% of their Chinese counterparts.

In terms of lipid intake, the Spanish population demonstrated significantly higher consumption, with only 33% meeting dietary recommendations, compared to 40.7% of the Chinese population. A previous study on the general Chinese population [43] revealed similar deficiencies in RDA compliance, although the protein intake (66 g/day) was lower than in our sample. These findings contrast with WHO reports [44], which indicate that older Chinese adults generally consume macronutrients below recommended levels. Further analyses of dietary intake among Chinese seniors have shown that fewer than a third meet recommended carbohydrate and fat intake levels, and fewer than a fifth consume adequate protein, with over half exceeding fat intake recommendations [45]. Consistently, studies of Chinese adults over 70 years old [46] have reported excessive protein and fat consumption, mirroring our findings.

A major distinction emerged in the lipid profile between Spanish and Chinese older adults. The Spanish group derived a lower percentage of calories from carbohydrates (43% vs. 62%) but a higher proportion from proteins (19% vs. 17%) and fats (36.7% vs. 19.8%). Differences in specific lipid components were also notable, with the Chinese group demonstrating a significantly lower saturated fat intake, with 93% meeting RDA guidelines. However, the polyunsaturated fat intake was lower in the Chinese group, whereas the monounsaturated fat intake was significantly higher among the Spanish group (44.4% higher), though neither group met the recommended intake. The polyunsaturated/saturated fat ratio and the total unsaturated/saturated fat ratio were more favorable in the Chinese group. The cholesterol intake also differed significantly, with 66.7% of Spanish participants meeting the recommendations, compared to only 40.7% of Chinese participants. A study on older Chinese individuals [47] yielded similar results, showing an inadequate energy and nutrient intake despite excessive fat consumption. Furthermore, nutritional surveys in other Asian countries [48] have highlighted the increasing reliance on saturated fats, declining intake of omega-3 polyunsaturated fats, and rising trans-fat and sugar consumption, trends also observed in this study.

In summary, the dietary imbalances observed in both populations reinforce the importance of culturally tailored nutritional strategies to promote better dietary habits among older adults engaging in physical activity. Future research should consider larger sample sizes and longitudinal designs to further explore the interplay between dietary intake, physical activity, and aging-related health outcomes.

This study presents certain limitations that should be considered when interpreting the results. One of the main challenges encountered was the difficulty in recruiting a sufficient sample of older Chinese adults, primarily due to cultural and language differences. These factors may have influenced both participation rates and the ability to ensure complete homogeneity between groups. As a result, the sample size of the Chinese group was smaller than that of the Spanish participants, which could limit the statistical power of the comparisons and the generalizability of the findings to the broader Chinese older adult population. Furthermore, the study focused exclusively on older adults, without including younger adult groups for comparison. Future research should aim to increase sample sizes and incorporate adult groups to assess whether differences in diet and physical activity are consistent across different age groups. This approach would help determine whether older adults deviate even further from the nutritional and physical activity requirements established for the general population. Expanding the scope of the study in this manner would provide a more comprehensive understanding of how aging, culture, and lifestyle factors interact in shaping dietary and physical activity patterns.

## 5. Conclusions

This study underscores the critical role of physical activity in shaping lifestyle and health outcomes in older adults. While no significant differences were found in the energy and macronutrient intake between active and sedentary individuals, the overall dietary patterns in all groups were inadequate, characterized by an excessive protein and fat intake alongside insufficient carbohydrate consumption. However, the greater engagement in lifelong physical activity observed in the Chinese population suggests that cultural factors play a key role in fostering a more health-conscious lifestyle.

Despite dietary imbalances in both Spanish and Chinese older adults, the latter demonstrated a better adherence to the carbohydrate and lipid recommendations, reflecting a more ingrained awareness of diet and physical activity as fundamental pillars of well-being. These findings highlight the urgent need for comprehensive interventions that integrate both regular exercise and proper nutrition to promote healthy aging. Encouraging the adoption of physical activity as a lifelong habit, akin to the Asian model, could significantly contribute to better health outcomes. Future research should expand sample sizes and compare different age groups to further explore how cultural influences shape dietary habits and physical activity engagement over the lifespan.

## Figures and Tables

**Figure 1 nutrients-17-01274-f001:**
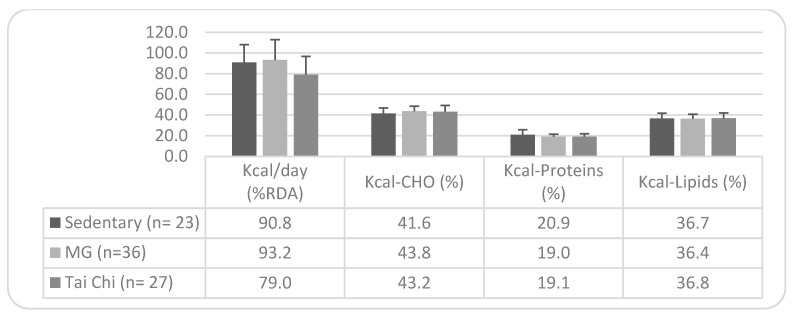
Caloric profile of the diet: differences by type of physical activity. Note. Mean values ± SD. n = sample size; SFAs = Saturated Fatty Acids; MUFAs = Monounsaturated Fatty Acids; PUFAs = Polyunsaturated Fatty Acids; and MG = Maintenance Gymnastics. Recommended targets: Energy contribution (% kcal) from SFAs = <7%; from MUFAs = 13–18%; and from PUFAs = <10%.

**Figure 2 nutrients-17-01274-f002:**
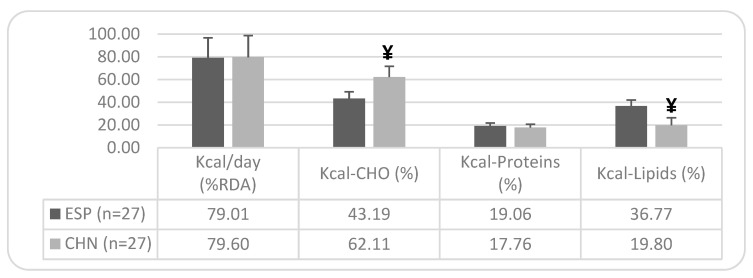
Caloric profile of the diet in the older Spanish and Chinese population: ethnic differences. Note: Mean values ± SD. Differences between Spanish and Chinese: ¥ = *p* < 0.05. n = sample size; RDA = Recommended Dietary Allowance; and CHO = Carbohydrates.

**Figure 3 nutrients-17-01274-f003:**
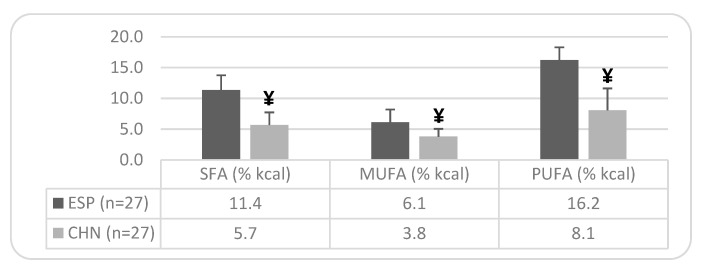
Caloric percentage of dietary fatty acids in the older Spanish and Chinese population: ethnic differences. Note: Mean values ± SD. Differences between Spanish and Chinese: ¥ = *p* < 0.05. n = sample size; SFAs = Saturated Fatty Acids; MUFAs = Monounsaturated Fatty Acids; PUFAs = Polyunsaturated Fatty Acids. Recommended targets: Energy (% kcal) from SFAs = <7%; from MUFAs = between 13–18%; and from PUFAs = <10%.

**Table 1 nutrients-17-01274-t001:** Descriptive of the sample of subjects according to their ethnicity.

Participants	ESP (n = 27)	CHN (n = 27)
Age	70.30 ± 4.94	67.59 ± 3.06
Weight (kg)	63.18 ± 11.84	64.68 ± 8.34
Height (cm)	159.03 ± 6.30	166.85 ± 6.49 #
BMI (kg/m^2^)	24.77 ± 3.60	23.19 ± 2.30

Note: Mean values ± SD. Level of significance: # = *p* < 0.05. n = sample size; BMI = Body Mass Index.

**Table 2 nutrients-17-01274-t002:** Descriptive of the sample of subjects according to the physical activity group.

Participants	Age	Weight (kg)	Height (cm)	BMI (kg/m^2^)
GM (n = 36)	71.25 ± 6.01	69.94 ± 9.15	158.55 ± 8.18 #	27.87 ± 3.7 ¥
TC (n = 27)	70.30 ± 4.94	63.18 ± 11.84	159.03 ± 6.30	24.77 ± 3.6
SD (n = 23)	78.91 ± 9.01	70.69 ± 13.16	164.30 ± 7.77	26.09 ± 4.08

Note: Mean values ± SD. Level of significance in differences of GM vs. SD: # = *p* < 0.05 and vs. TC: ¥ = *p* < 0.05. n = sample size; GM = Maintenance Gymnastics; TC = Tai Chi; SD = Sedentary.

**Table 3 nutrients-17-01274-t003:** Differences in energy intake and immediate beginnings in older Spanish adults according to type of physical activity.

SD (n = 23) GM (n = 36) Tai Chi (n = 27)	Population	Intake	RDA Compliance		
			Low (%)	Meets (%)	Exceeds (%)
Energy (Kcal/day)	Sedentary	1708.87 ± 299.02	26.09	69.57	4.35
	Maintenance Gymnastics	1820.11 ± 420.71	30.50	61.10	8.30
	Tai Chi	1629.44 ± 355.05	51.85	48.15	0.00
Proteins (%RDA)	Sedentary	194.21 ± 51.89	0.00	0.00	100.00
	Maintenance Gymnastics	199.29 ± 37.90	0.00	0.00	100.00
	Tai Chi	178.04 ± 39.78	0.00	3.70	96.30
			Compliance with Objectives		
			Low (%)	Meets (%)	Exceeds (%)
Carbohydrates (g)	Sedentary	165.71 ± 37.96	95.65	4.35	---
	Maintenance Gymnastics	184.12 ± 45.34	89.47	10.53	---
	Tai Chi	160.59 ± 39.19	77.78	22.22	---
Lipids (g)	Sedentary	70.20 ± 17.28		34.78	65.22
	Maintenance Gymnastics	74.68 ± 21.48	0.00	34.21	65.79
	Tai Chi	67.66 ± 18.75	0.00	33.33	66.67
Proteins (g)	Sedentary	88.96 ± 24.91	---	8.70	91.30
	Maintenance Gymnastics	85.36 ± 15.16	---	5.26	94.74
	Tai Chi	77.66 ± 18.52	---	3.70	96.30

Note: Mean values ± SD. Influence of physical activity type, where n = sample size; and RDA = Recommended Dietary Allowance. Compliance: Protein RDA between 10–15%. % = Percentage of the sample below, meeting, or exceeding RDA compliance (Low = between 0% and 80% RDA; Meets = between 81–120%; and Exceeds >120%) for the following daily energy intake targets: Proteins = 10–12%, Lipids < 35%, and Carbohydrates = 50–60%.

**Table 4 nutrients-17-01274-t004:** Differences in daily energy intake and macronutrients between older adults from Spain and China.

ESP n = 27 CHN n = 27	Population	Intake	RDA Compliance		
			Low (%)	Meets (%)	Exceeds (%)
Energy (Kcal/day)	ESP	1629.44 ± 355.05	51.85	48.15	0.00
	CHN	1804.74 ± 365.62	55.56	40.74	3.70
Proteins (%RDA)	ESP	178.04 ± 39.78	0.00	3.70	96.30
	CHN	167.43 ± 49.64	0.00	22.22	77.78
			Compliance with Objectives		
			Low (%)	Meets (%)	Exceeds (%)
Carbohydrates (g)	ESP	160.59 ± 39.19	77.78	22.22	---
	CHN	267.11 ± 64.11 ¥	7.41	92.59	---
Lipids (g)	ESP	67.66 ± 18.75	0.00	33.33	66.67
	CHN	40.39 ± 16.61 ¥	59.26	40.74	0.00
Proteins (g)	ESP	77.66 ± 18.52	---	3.70	96.30
	CHN	80.37 ± 19.64	---	11.11	85.19

Note: Mean values ± SD. Ethnicity influence. Differences between Spanish and Chinese: ¥ = *p* < 0.05. n = sample size; and RDA = Recommended Dietary Allowance. Compliance: RDA for proteins between 10–15%. % = Percentage of the sample by default and excess compliance with the RDA (Low = Between 0% and 80% of the RDA; Complies = Between 81% and 120%; and Exceeds > 120%) for the following daily energy targets: Proteins = 10–12%, Lipids: <35%; and Carbohydrates: 50–60%.

**Table 5 nutrients-17-01274-t005:** Differences in the lipid profile of the diet between older Spanish and Chinese individuals.

ES n = 27 CH n = 27	Population	Intake	RDA Compliance		
			Low (%)	Meets (%)	Exceeds (%)
SFA (g) (RDA en % Kcal)	ESP	20.90 ± 7.34	0.00	25.93	74.07
	CHN	11.95 ± 5.48 ¥	---	92.59	7.41
MUFA (g) (RDA en % Kcal)	ESP	29.54 ± 8.73	100.00	0.00	0.00
	CHN	16.44 ± 8.31 ¥	100.00	0.00	0.00
PUFA (g) (RDA en % Kcal)	ESP	11.03 ± 4.16	0.00	0.00	100.00
	CHN	7.63 ± 3.21 ¥	7.41	0.00	85.71
PUFA/SFA	ESP	0.56 ± 0.21	51.85	48.15	0.00
	CHN	0.74 ± 0.028 ¥	18.52	81.48	---
PUFA + MUFA/SFA	ESP	2.03 ± 0.42	55.56	44.44	0.00
	CHN	2.19 ± 0.54	33.33	66.67	---
Cholesterol (mg)	ESP	277.56 ± 95.29	0.00	66.67	33.33
	CHN	367.66 ± 164.16 ¥	---	40.74	59.26
Cholesterol (mg)/1000 Kcal	ESP	168.42 ± 42.56	0.00	7.41	92.59
	CHN	187.12 ± 96.08	---	14.81	85.19

Note: Mean values ± SD. Ethnicity influence. Differences between Spanish and Chinese: ¥ = *p* < 0.05. n = sample size; and RDA = Recommended Dietary Allowance. % = Percentage of the sample by default and excess compliance with the RDA (Low = Between 0% and 80% RDA; Complies = Between 81–120%; and Exceeds >120%) for the following lipid intake targets: Energy (% kcal) from SFAs = <7%; from MUFAs = between 13–18%; and from PUFAs = <10%. Compliance as a diet quality indicator: PUFA/SFA = >0.5; PUFA + MUFA/SFA = >2; Cholesterol = <300 mg; and Cholesterol (mg)/1000 kcal = <100.

## Data Availability

The original contributions presented in this study are included in the article. Further inquiries can be directed to the corresponding author.

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
