# Peer review of "Nutritional Profile Analysis of Active and Sedentary Older Adults: Differences Between Spain and China"

_nutrients, 2025, doi:10.3390/nu17071274_

Round 1

Reviewer 1 Report

Comments and Suggestions for Authors

Introduction:

Sedentary data is missing, you should add it

Methods:

It is very difficult understood how sample is, so, you have to explain better, are their from China or their are living in Spain. Where they are? how do you contact with all of them (From sport clubs?)

Physical activity instrument, by questionnaire? or only asking them? it is not sufficient.

Results:

descriptive sample table is necessary

table 3: there are some word in Spanish

figure 1: where is PTC people?

table 4 and sucessive: why only you have 27/27 participants? 

Reviewer 2 Report

Comments and Suggestions for Authors

Dear Authors,

After reading your manuscript, I have some questions/issues that need clarification listed below:

  1. Please rewrite the abstract. In the present shape, I can find no information about the results and conclusions;
  2. Please bear in mind that terms like “elder” or “elderly” evoke negative stereotypes of older adults, which can lead to othering older adults, bias against older adults, and poor outcomes for older adults. Instead of those terms, more neutral phrases are preferred, such as “older adult, “older person,” or “persons over 65.” Please consider using, for instance, “older adults” or “older patients” in your manuscript. I suspect that you are fully aware of it, because you mainly use other terms, and it happened only by accident (7 times in the whole paper);
  3. In the Introduction, you shed light on the situation of the Spanish population. As your work emphasized differences between Spain and China, please add corresponding information about the Chinese population (I mean statistics);
  4. Reading carefully the introduction, I feel it is too long. I know the information presented here is essential, but if it is possible to shorten it, please consider doing so. What is most important is that your introduction lacks a sentence summarizing the state of the art in the field of geriatric nutrition and the objective of your study.
  5. Please check your manuscript for typos (for instance, there is a lack of space here: “older adults.It is important”);
  6. Describing the methodology, you state:” To evaluate ethnic differences between Spanish and Chinese participants, an independent samples t-test was performed.” This is intriguing because only a quarter of the studied population are participants of Chinese origin. Please explain. Do you not think this group is too small to draw any conclusions?
  7. I cannot see the limitations of your study in the Discussion;
  8. The study is poorly discussed, I encourage reviewing the discussion carefully;
  9. Please restrict the conclusions to your study’s findings.

Best regards,

The reviewer.

Reviewer 3 Report

Comments and Suggestions for Authors

Evaluation of Manuscript Nutrients-3542505

This is a comparative study on the nutritional profiles in two countries, Spain and China. Below, I present my evaluation with the aim of assisting the authors.

Abstract: The abstract is incomplete, as it lacks the results and conclusion. I request the authors to use the Tables and Figures to present the main results and their statistics.

Introduction: For an experimental study, the introduction is too lengthy and reads more like a review than an original article. I suggest the authors rewrite it, focusing on presenting previous studies that highlight the importance of the topic, the gaps in prior research that this study aims to address, and the study's objectives and hypotheses. If the authors review previous studies published in *Nutrients*, they will notice that 3 to 5 paragraphs are sufficient for a well-written introduction. The introduction is crucial for readers to understand the study's purposes. In my understanding, the authors aimed to compare diets and physical activity levels in two countries with distinct cultures, but this should be more clearly articulated.

Methodology: Include the calculation of sample representativeness, and I believe it is important to present the participants' fitness levels prior to data collection. The procedures for measuring participants' diets are well-described, but the authors do not describe the methods for measuring physical activity levels. Tables 1 and 2 are unnecessary and should be converted into text. Table 3 is not fully translated into English, with some parts still in Spanish. Please revise this.

Results: Regarding the comparison of results, I believe the authors should focus their analyses on the study's objectives. If the intention is to compare elderly individuals from Spain and China in terms of physical activity and dietary habits, both sets of variables should be included in the same model rather than being analyzed separately as presented. Therefore, I suggest the authors redo the analyses.

Discussion: The discussion is well-written in parts, but the first paragraph only analyzes Spanish participants, while the next paragraph focuses on Chinese participants. Since this is a comparative study, the analysis should be joint, as the current approach contradicts the study's purpose and the presented results. Additionally, what is the ultimate goal of this study? If a nutrition professional reads this comparison, how can it be applied to their clinical practice? How can the data from this study be used to improve dietary procedures? I believe the study would be more complete if it included additional health data from the participants. Finally, I miss a discussion on the limitations of the study, future research perspectives, and practical applications. Please revise accordingly.

Round 2

Reviewer 2 Report

Comments and Suggestions for Authors

Dear Authors,

Thank you for your cooperation.

In my opinion, your manuscript, in its present shape, reflects the significance of your study much better.

Best regards,

The reviewer.

Reviewer 3 Report

Comments and Suggestions for Authors

The first version of the manuscript had many limitations, but the authors' efforts to improve the quality of the text are noticeable.